# Molecular Mechanisms of Alveolar Epithelial Stem Cell Senescence and Senescence-Associated Differentiation Disorders in Pulmonary Fibrosis

**DOI:** 10.3390/cells11050877

**Published:** 2022-03-03

**Authors:** Xiaojing Hong, Lihui Wang, Kexiong Zhang, Jun Liu, Jun-Ping Liu

**Affiliations:** 1Institute of Ageing Research, Hangzhou Normal University School of Medicine, Hangzhou 311121, China; xiaojing.hong@hznu.edu.cn (X.H.); wanglihui@hznu.edu.cn (L.W.); kxzhang@hznu.edu.cn (K.Z.); junliu262@hznu.edu.cn (J.L.); 2Department of Immunology and Pathology, Monash University Faculty of Medicine, Prahran, VIC 3181, Australia; 3Hudson Institute of Medical Research, Monash University Department of Molecular and Translational Science, Clayton, VIC 3168, Australia

**Keywords:** replicative senescence, DNA damage response, telomerase and telomeres, TGF-*β* signalling, pulmonary fibrosis, COVID-19

## Abstract

Pulmonary senescence is accelerated by unresolved DNA damage response, underpinning susceptibility to pulmonary fibrosis. Recently it was reported that the SARS-Cov-2 viral infection induces acute pulmonary epithelial senescence followed by fibrosis, although the mechanism remains unclear. Here, we examine roles of alveolar epithelial stem cell senescence and senescence-associated differentiation disorders in pulmonary fibrosis, exploring the mechanisms mediating and preventing pulmonary fibrogenic crisis. Notably, the TGF-*β* signalling pathway mediates alveolar epithelial stem cell senescence by mechanisms involving suppression of the *telomerase reverse transcriptase* gene in pulmonary fibrosis. Alternatively, telomere uncapping caused by stress-induced telomeric shelterin protein TPP1 degradation mediates DNA damage response, pulmonary senescence and fibrosis. However, targeted intervention of cellular senescence disrupts pulmonary remodelling and fibrosis by clearing senescent cells using senolytics or preventing senescence using telomere dysfunction inhibitor (TELODIN). Studies indicate that the development of senescence-associated differentiation disorders is reprogrammable and reversible by inhibiting stem cell replicative senescence in pulmonary fibrosis, providing a framework for targeted intervention of the molecular mechanisms of alveolar stem cell senescence and pulmonary fibrosis. **Abbreviations:** DPS, developmental programmed senescence; IPF, idiopathic pulmonary fibrosis; OIS, oncogene-induced replicative senescence; SADD, senescence-associated differentiation disorder; SALI, senescence-associated low-grade inflammation; SIPS, stress-induced premature senescence; TERC, telomerase RNA component; TERT, telomerase reverse transcriptase; TIFs, telomere dysfunction-induced foci; TIS, therapy-induced senescence; VIS, virus-induced senescence.

## 1. Introduction

Ageing, characterized by the gradual loss of the integrity of the body’s physiological function over time, occurs at a different pace as different tissue stem cells experience different stress pressures and damage accumulations. In the lung, pulmonary epithelial senescence represents the highest risk of idiopathic pulmonary fibrosis (IPF) [1,2]. Recently it was reported that SARS-Cov-2 viruses cause acute pulmonary virus-induced senescence (VIS), and subsequently fibrosis, illustrating a major mechanism of coronavirus disease 2019 (COVID-19) [3,4,5]. VIS features not only alveolar stem cell exhaustion, but also epithelial sloughing, dishevelled repair and fibrogenesis, ultimately resulting in recurrent dyspnea and respiratory failure similar to IPF [1,3,4,5,6,7,8]. As the alveolar epithelia contain predominantly alveolar monolayer squamous epithelial type 1 (AEC1) cells (80%) that are terminally differentiated without replicative capacity and the round cubic AEC2 stem cells (~15%) undertaking self-renewal, proliferation and differentiation to ACE1 [9,10], VIS exemplifies an acutely accelerated form of alveolar senescence in the wake of the cell cycle arrest [3]. However, the mechanism of the accelerated cellular replicative senescence remains to be elucidated in association with widespread alveolar epithelial interstitial damages, epithelial cell repair failure, myofibroblast fibrogenic proliferation and extracellular matrix deposition [3,4,5].

Differing from such pulmonary stem cells at the distal end of the bronchus as the clublike stem cells [11], distal airway stem cells (DASC) [11,12] and bronchioalveolar stem cells (BASC) [13], AEC2 stem cells reside within the alveolar epithelium with a significant heterogeneity with variable levels of the marker CD44, telomerase activity, and proliferative and differentiation potentials in response to a variety of cellular signalling in ageing [14,15]. Whereas chemically defined EGF/NOGGIN medium supports the basal proliferation of clonal AEC2 cells in the organoids [16], potentially through a constitutive activity of the single Wnt5a-signalling stromal fibroblast niche, an injury-associated activation of AEC2 stem cell expansion involves autocrine Wnt signalling associated with a potential checkpoint regulation in lieu of the niche juxtacrine Wnt signalling [17]. Responding to a transient interstitial macrophage-derived IL-1beta, the AEC2 stem cell population undergoes a reshuffle of a distinct subpopulation (damage-associated transient progenitors, DATPs) expressing Il1r1 for alveolar regeneration via a HIF1alpha-mediated glycolysis pathway [18]. However, in contrast to a full differentiation to AEC1 cells [16,18,19], AEC2 cell differentiation to AEC1 cells is hampered at an intermediate, partially differentiated status of AEC2 stem cells (called pre-alveolar type-1 transitional cell state, PATS or alveolar differentiation intermediate, ADI) under persistent inflammatory stress conditions, showing cellular replicative senescence in pulmonary fibrosis of both animal models and patients [4,18,19,20,21,22,23]. These findings of AEC2 stem cell senescence-associated differentiation disorder (SADD) may have significant implications in not only the impaired AEC1 cell replenishment during alveolar epithelial repair, but also aberrant trans-differentiation during the pathogenic development of pulmonary fibrosis (Figure 1).

Cellular replicative senescence occurs in various forms depending on the mechanisms leading to replicative senescence. Such forms have been described as stress-induced premature senescence (SIPS), oncogene-induced replicative senescence (OIS), developmental programmed senescence (DPS) and therapy-induced senescence (TIS), which display a general underlying mechanism of the permanent cell cycle arrest in cellular mitotic divisions. Initially referred to as the irreversible cell proliferation ability after a limited number of continuous population doublings for human diploid fibroblasts (HDFs), cell senescence occurs as a general feature of the different types of cell replicative senescence to the resistance to both cell proliferation and cell death signalling, e.g., without responding of mitosis to growth factor induction [24,25]. Thus, the primary cellular mechanism of cell replicative senescence converges on the cell cycle permanent arrest provoked by significant stress insults along with sustained DNA damage in response to a variety of physical, chemical and biological stimuli (Figure 1). Although a variety of diverse stressors accelerate premature terminations of the cell cycle, including chemical toxins, antibiotics and oxidative stress in the airways [24,26,27,28,29], cell replicative senescence appears irreversible as demonstrated in SIPS and OIS [30]. With the enhanced mitogenic signalling such as activation of the Ras/MAPK pathway, and the epigenetic modifications such as histone-3 lysine-9 trimethylation (H3K9Me3) in the senescence-associated heterochromatin foci [30,31], OIS occurs to serve as an oncogenic checkpoint to prevent tumor cell clonal expansion and micro-evolutionary transformation with increased oncogenic proteins (such as c-myc [32] and mTOR [33]). In contrast to OIS, SIPS and VIS occur with excessive damages to serve as a proliferative and differentiative checkpoint to prevent cellular damage from passing onto daughter cells [3,34,35]. However, it remains unclear how the cell cycle arrest is specifically regulated in SIPS, OIS and VIS. Previous studies suggest that cellular replicative senescence involves the activation of retinoblastoma protein (RB) signal transduction pathways of tumor suppressor genes *p53-p21^CIP1^*/*CDKN1A* and *p16^INK4A^*/*CDKN2A* [3,36,37,38]. Additionally, DPS may occur via the transforming growth factor-*β* (TGF-*β*), PI3K and ERK1/2 pathways to regulate tissue remodelling [39]. Moreover, the ER unfolded protein response (ER^UPR^) may participate in TIS with increased ER associated degradation (ERAD), resulting in ER-related senescence-associated apoptosis in a caspase-12 and caspase-3 dependent manner [40]. While it remains unclear if ER-mediated TIS entails DNA damage response (DDR), recent studies show that TIS reprograms cancerous cells to acquire the cellular phenotypic features of cellular stemness [41].

Intriguingly, the intrinsic reprograming mechanism of a potential cell transition from ceasing replication to entering senescence may involve the transcriptional repression of the *telomerase reverse transcriptase* (*TERT*) gene expression, conferring cell replicative senescence on oncogene-stimulated cell division [42]. Additionally, TP53 phosphorylation at S15 residue [38] and TP53 fluctuation in oscillatory dynamics [43] may also be involved in the pathway switching from cell replicative senescence to a cell proliferative state. Furthermore, DNA methylation of the Oct4A enhancers may regulate the TP53-dependent TIS through tuning alternative splicing [44]. With an instrumental non-cell-autonomous role of SASP [41,45,46], cell replicative senescence plasticity represents a hotspot of further investigation in terms of autocrine/juxtacrine/paracrine actions of growth factors, proteases, cytokines, chemokines and extracellular matrix (ECM) components [47,48,49,50,51]. In this regard, studies showed that the extracellular vesicles (EVs)—exosomes, ectosomes, microvessels and microparticles—are increased in cell replicative senescence responding to ionizing radiation, chemical reagents or overexpression of oncogenes in a number of cell types including epithelial cells [52,53], and that TP53 activation increases EV release and mediates prostate cancer cell senescence induced by telomere shortening or radiation [53], suggesting that extracellular vesicular signalling represents a potential mechanism shaping cellular responses to and from replicative senescence. In the following sections, we focus on the phenotypic characteristics of SADD in the development of pathological fibrogenic remodelling and explore the molecular mechanisms mediating telomere maintenance, homeostatic disruption and prophylactic/therapeutic interventions as well in stress-induced cell replicative senescence and pulmonary fibrosis.

## 2. Senescence-Associated Differentiation Disorder (SADD)

As the lung peripheral tissue stem cells that are essentially required for alveolar epithelial damage repair and regeneration [10,11,12,13], AEC2 stem cells are compromised in replenishing AEC1 cells in pulmonary fibrosis [23,54,55]. In a long-term process of AEC2 cell self-renewal and directional differentiation to form AEC1 cells by pedigree tracing [10], AEC2 cells respond to chronic stress with the phenotypes of permanent termination of the cell cycle and differentiation to AEC1 [9,18,19,22] by an unclear molecular mechanism [18,19,20,21,22,23]. In the case of diffuse alveolar damage, in addition to discontinued differentiation to AEC1 cells, AEC2 stem cells arrested at G_2_/M of the cell cycle can express a high level of KRT8, and exhibit the cubic and partial spreading morphologies [18,19,20,21,23,56,57]. Moreover, the AEC2 senescent cells appeared to be undergoing a program of transdifferentiation to a mesenchymal state in association with increased levels of α-SMA and collagen [35,58,59,60,61,62,63]. Collectively, these studies implicate AEC2 senescent stem cells as behaving with a SADD characteristic of the loss of differentiation to AEC1 cells and the gain of aberrant transdifferentiation during pulmonary pathogenic fibrogenesis. The features of stress-induced AEC2 stem cell replicative senescence and SADD in mice are consistent with recent clinical findings that AEC2 stem cells express increased senescence markers, including p16^INK4A^/CDKN2A, p14^INK4B^/CDKN2B, TP53 and p21^CIP1^/CDKN1A [3,4,7], and that AEC2 stem cell SADD existed throughout the course studied in the pneumonia-induced acute respiratory distress syndrome and pulmonary fibrogenesis [4,64,65]. Importantly, AEC2 stem cell SADD manifested in association with pulmonary interstitial deposition in both COVID-19 [64] and IPF [66]. It is thereby probable that AEC2 stem cell replicative senescence mediates the differentiation failure of AEC2 stem cell replenishing damaged AEC1 cells by a mechanism at least involving the compromised proliferative potential [3]. Inducing AEC2 stem cell senescence or otherwise depletion demonstrated that AEC2 senescence, rather than AEC2 cell loss, promotes progressive fibrosis [38], consistent with a gain of malfunction of senescent AEC2 cells as a mechanism in causing pathogenic fibrogenesis. Moreover, preventing AEC2 stem cells from entering replicative senescence [35], or eliminating replicative senescence cells in COVID-19 [3], rescues pulmonary fibrosis in the animal models, underscoring that AEC2 stem cell replicative senescence is a prerequisite driver of pulmonary fibrosis [38].The mechanism of AEC2 stem cell SADD remains to be investigated extensively, although dysregulation of several intracellular signalling pathways (including the TGF-β family) participates in the deregulation of AEC2 differentiation [19,67]. Notably, serving as a regulatory switch, BMP4 was reported to inhibit AEC2 stem cell proliferation while stimulating AEC2 stem cell differentiation to AEC1 cells [67]. Conversely, the antagonists, follistatin and noggin, promote AEC2 stem cell self-renewal and inhibit AEC2 stem cell differentiation to AEC1 cells [67]. Moreover, a number of other extracellular factors including TGF-β, TGF-α, IL-13, TNF-α, IL-1β, IL-1R1, retinoic acid and platelet-derived growth factor are likewise implicated in shutting down AEC2-to-AEC1 cell differentiation in the milieu of the alveolar interstitium [18,68,69]. It is suggested that TGF-β and follistatin-like 1 form a positive feedback loop in accelerating the pathogenic differentiation of fibrogenesis in IPF [70]. While the downstream effectors of TGF-β may engage in regulating the telomerase *TERT* gene and telomere maintenance negatively, it has been shown recently that protecting telomeres from dysfunction eliminates stress-induced AEC2 stem cell senescence and subsequent pulmonary fibrosis (see below for details).

In addition to the discontinued AEC2-to-AEC1 cell differentiation in SADD of IPF, AEC2 cell transdifferentiation in the pathological fibrogenic condition [35,58,59,60,61,62,63] may involve epithelial-mesenchymal transition (EMT) [62,63,71,72,73,74,75,76]. Evidence of AEC2 cell contributing to fibrogenesis via transdifferentiation includes co-localization of SPC with α-SMA, Col1α1 and hydroxyproline in the mouse models of pulmonary fibrosis [35,61,62,63]. In addition, recent single-cell RNA sequencing studies revealed that a cell population of replicative senescence is associated with transdifferentiation in the profiles of KRT5^−^/KRT17^+^ marks and increased gene expressions of a subset of mesenchymal genes such as Col1α1 in the fibrotic lung of patients [56,57]. Lineage-tracing studies in mice show that AEC2 stem cells undergo aberrant cellular remodeling with stretching morphological alterations in association with replicative senescence characteristic of SADD, by a mechanism involving increased TGF-*β* and TP53 signaling [19,38,77]. Thus, it is conceivable that SADD underlies the discontinuation of AEC2 stem cell differentiation to AEC1 cells and initiations of myofibroblast activation and matrix deposition under the conditions of alveolar senescence-associated low-grade inflammation (SALI) [78,79,80,81] with inflammatory cell and cytokine signaling [19,62,63,71,72,73,74,75,76,82,83]. Further studies are required to investigate if cell replicative senescence may dictate SADD through intracellular mechanisms prior to SALI (Figure 2), so inhibiting cell replicative senescence may impede SADD as to prevent fibrogenesis in the animal models of telomere dysfunction-induced pulmonary fibrosis [35].

## 3. Telomere Dysfunction Mediates Pulmonary Senescence and Fibrosis

AEC2 stem cell replicative senescence has been shown to involve DDR in SADD and IPF [35,84,85,86,87] under SALI, which is subserved by senescence-associated secretory phenotype (SASP) and infiltrations of inflammatory and immune cells [78,79,80,81] (Figure 1 and Figure 2). Genetically, loss of function mutations of the genes encoding *TERT*, *telomerase RNA component* (*TERC*), *poly-(A)-specific ribonuclease* (*PARN*) and *regulator of telomere length 1* (*RTEL1*) underpins patient familial recessive hereditary IPF [2,18,88,89,90,91,92,93]. Notably, over a third of IPF cases with hereditary susceptibility of autosomal recessive gene damages could be related to telomere-related gene mutations including telomerase catalytic subunit *TERT* and RNA subunit *TERC* genes [6,35,93,94,95,96,97], and some could be related to mutations of the genes coding SPC or SPA of alveolar stem cells [98,99,100]. With the hypothesis that genetic susceptibility involves vulnerable genetic elements such as telomeres predisposing increased pulmonary sensitivity to environmental hazard-accelerated damages [35], it has been shown that stress-induced DDR occurs rapidly to telomeres, in a more abundant scale in telomeres as compared with non-telomeric regions in the genome [35,101,102]. The telomeric DDR takes place irrespective of telomerase activity, resistant to DNA repair, rendering irreparable DDR to be persistent in causing cellular replicative senescence [103,104,105,106]. Thus, preventing stress-induced telomere damages may protect the cells from entering replicative senescence and fibrogenesis.

Environmental factors triggering pulmonary senescence and fibrotic disease onset have been a hot topic in aiming to prevent the incurable disease IPF without yet effective therapy [91,96]. Since the 1980s, environmental stress factors that drive pulmonary fibrotic onset include smoking, bacterial toxin bleomycin [107,108], ionizing radiation (IR) [109,110,111], and oxidative metabolite reactive oxygen species [84,112,113,114,115]. Such environmental stress cues as bleomycin [58,116,117], IR [105,118,119] or H_2_O_2_ [102,120,121,122] have been shown to induce telomere DNA damage, resulting in telomere shortening. Consistent with cigarette smoking as causing pulmonary lesions by a mechanism of telomeric DDR [123,124,125], a significant number of smokers with IPF and chronic obstructive pulmonary disease (COPD) demonstrated acceleratory shortening of telomeres [47,123,126]. We demonstrated that radiation exposure, oxidative stress (such as H_2_O_2_) or bacterial product bleomycin each trigger telomere shelterin protein TPP1 degradation in AEC2 stem cells, provoking telomere uncapping, DDR, stem cell exhaustion, fibrogenic gene expressions, and pulmonary fibrosis [35]. These findings indicate a primary mechanism through which environmental stress induces AEC1 cell damage and coerces AEC2 stem cells to succumb to replicative senescence and SADD in pulmonary fibrosis.

In the telomere syndrome traits, IPF appeared more common than bone marrow diseases [58,95,127] (such as aplastic anemia [128] or dyskeratosis congenita (DC) [129,130,131,132]). The incidence of IPF is related to the aggravation of lung epithelial injuries, alveolar stem cell replicative senescence and SADD induced by environmental stress factors such as smoking, infection, radiation injury, oxidative stress [16,35,93,117]. It is thought that once in replicative senescence, AEC2 stem cells are entangled in the milieu of SALI in relaying bidirectional signal transductions via a variety of inflammatory factors to and from senescent cells, thereby driving the development of fibrogenesis [78,133,134,135]. Animal studies have demonstrated that environmental stresses, such as viral infection, induce extensive inflammatory responses, interstitial hyperplasia and edema of the lung tissue, resulting in fibrosis and dyspnea [35,136,137], characteristic of chronic pulmonary epithelial reduction, diffuse alveolitis and interstitial myofibroblast proliferation [78,138]. Investigation of the key molecular mechanisms mediating stress-induced pulmonary cellular senescence may unveil molecular targets pivotal for intervention in pulmonary fibrosis as in IPF. What remains unclear for decades includes how telomere shortening or replicative senescence is triggered and accelerated to provoke IPF by various environmental stressors. Telomeres are composed of the DNA repeats, (TTAGGG)_n_, and their binding protein complex shelterin at chromosome ends, underpinning chromosome integrity and stability in the regulation of genome homeostasis in cell development, proliferation and differentiation [139,140,141,142]. Shelterin contains six proteins, telomere repeat binding factor 1 (TRF1), TRF2, RAP1, TIN2, POT1 and TPP1, and covers both double-stranded and single-stranded telomeric DNA, preventing telomeres from inadvertently activating DDR and replicative senescence in stress responses [143,144]. Under physiological conditions, telomere maintenance, or shortening as it occurs with cell replication, is regulated by the levels of shelterin and operates as the mitotic clock that results in their exit from the cell cycle in replicative senescence [145,146]. However, telomere dysfunction occurs as a function of sustained abundance of environmental stress stimulations, preceding non-telomeric DNA damages which ensue to telomere dysfunction [35]. Critical stress pressures [6,35,119,147,148], and chronic insufficiency of telomeric DNA repair, such as repressions of telomerase genes, repressions [35,78,149,150], together result in telomere shortening, ultimately premature senescence, and ageing-related diseases. Telomere dysfunction as a potential cause of ageing-related diseases [127,151,152] results from mutations of telomere related genes, including those encoding shelterin proteins [143], and telomerase [78,153].

Shelterin and telomerase function to protect telomeres by preventing cells from telomere DDR [29,154,155,156,157] and entering senescence [27,28,78,152,153]. Genetic mutagenesis of TRF2 or telomerase genes induces pulmonary fibrosis [58,78], whereas transgenically expressing *TERT* holds a therapeutic potential in alleviating pulmonary fibrosis [158]. In addition, the *ACD* gene coding for TPP1, which caps the telomere ends and recruits telomerase [159,160,161,162,163,164,165], was associated with such stem cell diseases as aplastic anemia with bone marrow failure [166] and Hoyeraal–Hreidarsson syndrome [167], and the expression levels of TPP1 were significantly reduced in the lung tissue of patients with chronic obstructive pulmonary disease (COPD) [168]. These studies suggest that qualitative and quantitative alterations in telomere shelterin and telomerase components accelerate telomere shortening, stem cell replicative senescence and related disorders.

### 3.1. Telomerase Gene Deficiency in Ageing-Related Disorders

Telomerase replenishes telomeres to counteract shortening by *TERC* template reverse transcription [169,170]. Telomerase catalytic subunit TERT determines telomerase activity in lengthening telomeres, conferring the ability of continuous proliferation and survival on stem cells and tissue precursor cells [171,172,173,174,175,176]. In addition to elongating telomeres, TERT is involved in maintaining telomere heterochromatin and synthesizing double-stranded RNA through RNA-dependent RNA polymerase activity (RdRP) [177,178,179]. Telomerase deficiency due to *TERT* mutations results in stem cell replicative senescence and exhaustion, tissue fibrosis, aplastic anemia, and skin diseases of premature ageing [35,78,132,180,181]. While *TERT* gene expressions are fundamental to telomerase activity and stem cell renewal, tissue regeneration, the molecular mechanisms underlying *TERT* gene expression remain largely elusive [182]. Multiple lines of evidence indicate that several transcription factors play fundamental roles in regulating *TERT* transcription [183] (Figure 3 and Figure 4). Our laboratory established for the first time that downstream MAP kinase signalling [184], ETS transcription factors bind to the *TERT* gene promoter ETS binding motif (*CCTT*) directly, functioning as an essentially key transcription factor for *TERT* gene expression in cancer [184,185]. Moreover, ETS activates *TERT* transcription not only directly by binding to the *CCTT* element and in cooperation with the proto-oncogene c-myc that binds to the E-box (*CACGTG*) at the *TERT* promoter in the regulation of telomerase activity and cell proliferation [185,186,187,188] (Figure 3 and Figure 4). The role of ETS in *TERT* gene expression has been confirmed independently in cancer patients with C→T nucleotide mutations creating additional ETS binding sites [189,190] and mechanistically [191]. Furthermore, Wu et al. showed first that c-myc activates *TERT* by binding to the E-box at the *TERT* gene promoter [192]. However, the action of c-myc at the E-box of *TERT* promoter is regulated negatively by Max/Mad1 interaction [193] and by TGF-β signalling via Smad3 interactions with c-myc and the *TERT* gene promoter [186,188] (Figure 3 and Figure 4). Our laboratory demonstrated that estrogen upregulates *TERT* gene expression and telomerase activity in the endocrine organs of ovary and adrenal gland for tissue homeostasis [149,150,194,195]. Deficiency of estrogen in mice by targeted disruption of the aromatase gene responsible for estrogen synthesis results in compromised cell proliferation and stem cell exhaustion [149,150]. Interestingly, hormone replacement therapy increases the activities of telomerase and stem cell renewal positive for Ki67 staining and the stem cell markers stem-cell antigen-1 (Sca-1) and c-kit expression, attenuating premature ageing and tissue atrophy at both adrenal and ovarian glands [149,150]. These studies are consistent with the current notion that telomerase TERT is required for the maintenance of telomeres and stem cell functionalities in the endocrine, epithelial and endothelial tissues against tissue degenerative changes and proliferative disorders [196,197,198].

### 3.2. TGF-β Signaling to Telomerase TERT Gene Repression

TGF-β signaling is closely involved in pulmonary fibrosis [19,20,22,23,56,57,82,199,200,201]. The mechanisms of how TGF-*β* family cytokines regulate the intracellular events mediating pulmonary fibrogenesis remain unclear [202]. Studies indicate that mediating the interactions between AEC2 stem cells and fibroblasts, TGF-*β* released from cells [203] induces Sonic Hedgehog (SHH) pathway in alveolar epithelial cells by autocrine mechanisms, whereas SHH induces TGF-*β* in lung fibroblasts stimulating myofibrosis by paracrine mechanisms, suggesting that re-emergence of SHH in epithelial cells mediates TGF-*β* signaling and induces myofibroblast differentiation in a Smoothened receptor-dependent manner with subsequent transcription factor Gli1 activation of the *α-SMA* promoter [204]. In addition, elevated extracellular mechanical tension between myoblasts and AEC2 stem cells activates TGF-*β* RII receptor signaling in AEC2 stem cells, resulting in AEC2 stem cells with increased TGF-*β*1 and becoming gradually unable to differentiate to AEC1 cells [23]. Deletion of the TGF-*β* RII receptor specifically in lung epithelium protects mice from bleomycin-induced pulmonary fibrosis, further supporting a central role TGF-*β* signaling of alveolar epithelium in fibrogenesis [82,205]. Reflecting SADD, moreover, recent studies show that in radiation-induced pulmonary fibrosis, both TGF-*β* and GSK3*β* are increased in AEC2 cells undergoing transdifferentiation with increased AEC1 cells and mesenchymal markers such as α-SMA [206]. These studies suggest that TGF-*β* in the intercellular milieu stimulates AEC2 cell transdifferentiation by mechanisms involving GSK3*β* in association with SHH cellular signaling.

The mechanism of intracellular transduction initiated by TGF-β in the induction of pulmonary fibrosis entails receptor-mediated Smad3 signaling. Firstly, epithelium-specific disruption of TGF-*β* RII receptor increases Smad2 phosphorylation and decreases Smad3 phosphorylation, and protects mice from bleomycin-induced pulmonary fibrosis with increased survival [82] (Figure 3). In addition, knockout of Smad3 [207], or inhibition of phosphorylated Smad3 into the nucleus by polypeptide [208], inhibits TGF-β1-induced pulmonary fibrosis in mice. Moreover, in an attempt to determine the downstream mechanisms of TGF-*β* activation of Smad3 co-transcription factor, our laboratory demonstrated that by acting on different specific RII receptors, TGF-*β*1 induces Smad3 translocation from the cytoplasm to the nucleus where Smad3 binds the *CAGA* box in the *TERT* gene promoter interfering c-myc binding and repressing *TERT* transcription [188,209,210,211] (Figure 3 and Figure 4). Furthermore, the TGF-*β* family member BMP7 induces the *TERT* gene repression in a BMP RII receptor- and Smad3-dependent manner [211,212]. Chronic exposure to BMP7 results in telomere shortening, cell replicative senescence and apoptosis, but mutation of the BMP RII receptor (but not TGFbRII, ACTRIIA or ACTRIIB receptors) inhibits BMP7-induced *TERT* repression, leading to increased telomerase activity, lengthened telomeres and continued cell proliferation [211]. The effect of BMP7-induced cell replicative senescence and apoptosis is *TERT* repression-dependent as overexpression of *TERT* reverses BMP7-induced cell replicative senescence [211]. These data together suggest that Smad3-mediated repression of the *TERT* gene and shortening of telomeres are critical to TGF-β induced pulmonary senescence, SADD, and pathological fibrogenesis.

### 3.3. Stress-Induced TPP1 Degradation and Telomere Uncapping

In the regulation of telomerase lengthening of telomeres, recruiting telomerase to telomeres represents a most effective step in telomere maintenance. Our recent studies show an intertwined relationship of TPP1 capping of telomeres, recruitment of telomerase, deficiency-induced telomere uncapping, pulmonary senescence, and fibrosis [35,213]. By targeted protection of telomere uncapping, we demonstrate that pulmonary senescence and fibrosis with dyspnea are altogether prevented in mice under persistent fibrogenic stresses including whole body ionizing radiation or pulmonary bacterial toxin bleomycin [35]. The stress-induced pulmonary senescence and fibrotic onset are targetable through animal inhalation of either recombinant cDNA coding for the telomere protein TPP1 or a small 8-mer peptide (telomere dysfunction inhibitor or TELODIN) to prevent telomere uncapping, indicating an early effective intervention strategy on the mechanisms of telomere dysfunction and subsequent cellular senescence prior to differentiation disorder of pulmonary fibrosis [35,213]. These findings dovetail with a large body of literature indicating that telomere DDR underpins IPF [35,58,93,95,103,104,105,106,214], providing an important strategy for intervention of cell replicative senescence-associated diseases.

TPP1 is a subunit of shelterin that is a multi-subunit protein complex capping telomeric DNA [143,144]. We showed that pulmonary fibrosis initiated by chronic stress in mouse models is mediated by the tumor suppressor protein FBW7-mediated degradation of TPP1 [35]. FBW7 E3 ubiquitin ligase binds TPP1 and mediates stress-stimulated TPP1 multisite polyubiquitination at K299, K453 and K459, causing TPP1 degradation in proteasomes and telomere uncapping [35]. The multisite polyubiquitination of TPP1 requires FBW7 interaction with a Cdc4 phosphodegron (CPD) site [215,216] in the serine/threonine (S/T) region (aa 341–482) of TPP1, with the CPD being phosphorylated by GSK3β at the S354 and S358 residues [35]. We showed that the binding to the phosphorylated CPD of TPP1 by FBW7 is mediated by the seventh *β*-strand blade containing the ^689^R residue frequently mutated in cancers at the WD40 propeller domain of the C-terminal region of FBW7 [35]. Consistent with stress-mediated TPP1 degradation, thereby telomere shortening and pulmonary fibrosis, we found that under-expression of TPP1 is sufficient to induce pulmonary senescence and fibrosis, indicating that TPP1 polyubiquitination at multiple sites triggered by stress incurs telomere uncapping, DDR and shortening, resulting in pulmonary senescence and fibrosis [35]. Moreover, overexpression of TPP1 enhances respiratory physiological function with increased AEC2 stem cell population with lengthened telomeres, and confers pulmonary resistance to stress-induced onset of pulmonary fibrosis [35].

## 4. Targeted Intervention of Cellular Senescence and Tissue Fibrosis

Since cellular replicative senescence plays a causal role in tissue remodelling, it appears particularly appealing to determine if elimination or prevention of cellular replicative senescence may provide beneficial outcomes to mitigate pathogenic tissue remodelling and ageing-related disorders under various disease conditions. Considerable evidence suggests that targeted removal or prevention of cell replicative senescence alleviates SADD in fibrosis, including VIS or stress-induced fibrosis [3,35].

### 4.1. Targeting Anti-Apoptotic Gene Bcl-2 to Clear Senescent Cells

Since replicative senescence cells have upregulated anti-apoptotic proteins Bcl-w and Bcl XL which underpin senescent cells’ antiapoptotic ability with long-term survival, drugs that inhibit Bcl survival proteins have been studied and named as senolytics for removing replicative senescence cells [217,218] (Table 1). Although the U.S. Food and Drug Administration (FDA) approved the selective Bcl-2 inhibitor venetoclax (abt-199) used in leukemia, it has not had a significant effect on anti-ageing tests in vitro. Compound screening-identified that its homologue navitoclax (abt-263) effectively inhibits the effects of Bcl-2, Bcl XL and Bcl-w, and induces apoptosis and thus clearance of replicative senescence cells [217]. In addition to abt-263, siRNAs and chemical inhibitor abt-737 simultaneously inhibit Bcl-w and Bcl XL, induce replicative senescence cell specific apoptosis, reduce RC in the populations of hematopoietic stem cells, muscle stem cells, and pulmonary and epidermal tissues of mice, with cleared replicative senescence cells harboring DNA damage and p14ARF-activated TP53 [218]. Moreover, a combination of dasatinib and quercetin (D + Q) reduces senescent cells, improving mouse idiopathic pulmonary fibrosis [219], and nerve regeneration and obesity-related anxiety behavior [220].

Recently, the use of navitoclax and D+Q selectively eliminated VIS cells and mitigated COVID-19-reminiscent lung disease with reduced inflammation in SARS-CoV-2-driven hamster and mouse models [3]. Treatment with navitoclax for less than a week to the SARS-CoV-2 infected Syrian golden hamster model with increased p16^INK4A^ led to a profound decrease in senescent cells and SALI [3]. On another hamster model of the Roborovski dwarf animals, the navitoclax or D + Q regimen led to a substantial reduction of the cell senescence markers H3K9me3 and p16^INK4A^ in the respiratory epithelium and increased lifespan [3]. Furthermore, in two randomized clinical trials (NCT04578158 and NCT04861298), quercetin showed significant effects of symptom improvements with significant risk reductions regarding the needs of hospitalization and oxygen therapy in a total of 194 COVID-19 patients [3]. The senolytic targeting of VIS as a novel regimen option against COVID-19 indicates that cellular replicative senescence is causative to severe stress-induced acute SADD, tissue fibrogenic remodelling, and phenotypic onset, and that eliminating cellular replicative senescence is beneficial in managing pulmonary fibrosis.

### 4.2. Targeting TP53 and p16^INK4A^ Tumor Suppressor Genes to Clear Senescent Cells

Recent studies have consistently shown that senescent cells can be eliminated by targeting the cells with high levels of p16^INK4A^ and TP53 tumor suppressor gene expressions. Because p16^INK4a^ is significantly increased in senescent cells, AP20187 that targets FK506 using a minimal p16^INK4a^ promoter element triggers a dimer formation of FK506 binding protein and caspase-8 to effectively induce senescent cell apoptosis, resulting in a significant reduction of the incidence and mortality of cardiovascular diseases in aged mice [221]. This regimen of p16^INK4a^ sensitive caspase-8-mediated apoptosis also delays osteoporosis and persistent intervertebral disc proteoglycans [217], restores nerve regeneration in obese induced by high-fat feeding or leptin receptor deficiency, and reduces anxiety-related behaviors in mice [220]. In another model of p16^INK4A^-3MR transgenic mice, the expression of a reporter protein (3MR) under the control of p16^INK4A^ to selectively remove senescent cells through using the small molecule compound UBX0101 reduced post-traumatic osteoarthritis and extended lifespan [222]. Furthermore, studies show that TP53 is crucial in AEC2 stem cell senescence preceding pulmonary fibrosis, especially with potential accumulation of phosphorylated TP53 at the S15 residue [38,77,223]. Therefore, the use of a FOXO4 polypeptide—harboring 60 amino acids to interfere with the interaction between FOXO4 and TP53—may be fundamental in triggering TP53-dependent apoptosis of senescent cells, delaying fibrogenesis and accelerating the rehabilitation of pulmonary ageing [223].

**Table 1 cells-11-00877-t001:** Senescence-Targeting Senolytics and Other Compounds.

Class	Senolytic Agent	Mechanism of Action	Reference
BCL-2 Family Inhibitors	Dasatinib + Quercetin ABT-737 ABT-263 A-1331852 A-1155463	PI3K/Akt inhibition/DNA intercalation Inhibiting BCL-2, BCL-XL, BCL-W Inhibiting BCL-2, BCL-XL, BCL-W Inhibiting BCL-XL Inhibiting BCL-XL	[3,217,218,220,224]
Targeting p53	FOXO4-DRI UBX0101 RG7112 (RO5045337) P5091 P22077 NOX1 and NOX2 dual inhibitor PAI-1 inhibitor (TM5275)	Disrupting FOXO4-p53 interaction Disrupting MDM2-p53 interaction Disrupting MDM2-p53 interaction USP7 inhibitor USP7 inhibitor Activating p53 and apoptosis Activating p53 and apoptosis	[222,223,225,226]
HSP90 Inhibitors	17-DMAG (alvespimycin) Geldanamycin 17-AAG (tanespimycin) Ganetespib	Disrupting HSP90-AKT interaction Attenuating HSP90 Attenuating HSP90 Attenuating HSP90	[227,228]
Natural Products and their Analogues	Fisetin Curcumin *O*-Vanillin (curcumin metabolite) EF-24 (curcumin analogue) Piperlongumine and its analogues (compounds 47–49) GL-V9	BCL-2, PI3K/AKT, p53, NF-κB and more Down-regulating Nrf2 and NF-κB pathways Down-regulating Nrf2 and NF-κB pathways Attenuating the BCL-2 family Unclear, OXR1, NF-κB - Unclear, increasing ROS, alkalizing lysosome	[224,229,230,231,232]
Cardiac Glycosides	Proscillaridin A Ouabain Ouabagenin Digoxin Bufalin K-Stropanthin Strophanthidin	Inhibiting Na^+^/K^+^-ATPase Inhibiting Na^+^/K^+^-ATPase, increasing BCL2-Family protein NOXA Inhibiting Na^+^/K^+^-ATPase Inhibiting Na^+^/K^+^-ATPase, increasing BCL2-Family protein NOXA Inhibiting Na^+^/K^+^-ATPase Inhibiting Na^+^/K^+^-ATPase Inhibiting Na^+^/K^+^-ATPase	[233,234]
Galactose Modified Prodrugs	SSK1 Pro-drug A (JHB75B) Nav-Gal 5FURGa	Targeting SA-β-galactosidase Targeting SA-β-galactosidase Targeting SA-β-galactosidase Targeting SA-β-galactosidase	[235,236,237]
PROTACs	PZ15227 ARV825	Degrading BCL-XL Degrading BRD4	[238,239]
Miscellaneous	Fenofibrate Azithromycin Roxithromycin Tamatinib (R406) MitoTam Panobinostat AT-406 Rapamycin Metformin	PPARα agonist Inducing autophagy and glycolysis NOX4 Syk inhibitor, FAK and p38MAPK Reducing mitochondrial membrane potential, Inhibiting OXPHOS Histone deacetylase inhibitor Inhibitor of c-IAP1, c-IAP2 and XIAP Inhibiting mTOR, p16 and p21 Inhibiting NF-κB pathways/AKT	[240,241,242,243,244,245,246,247]

### 4.3. Preventing Telomere Dysfunction and Pulmonary Fibrosis by TELODIN

By screening a peptide library, we discovered an 8-mer peptide (TELODIN) that significantly inhibited telomere dysfunction [213]. Corresponding to the *β*-turn hairpin-like blade 7 of FBW7 E3 ubiquitin ligase WD40 domain, as synthesized in a native or retro-inverso (reversed, inversed in dextral amino acids) configuration, TELODIN competitively inhibited stress-induced TPP1 accelerated turnover, telomere shortening and pulmonary fibrosis once applied through intratracheal instillation in mice [35,213]. TELODIN inhalation through the respiratory airway promoted alveolar stem cell proliferation and enhanced pulmonary resistance to stress-induced pulmonary fibrogenic onset induced by different chronic stresses including ionizing radiation, or by overexpression of FBW7 in causing TPP1 deficiency, telomere uncapping and DDR [35]. The effect of TELODIN on alveolar stem cell proliferation is due to increased TPP1 and subsequently protection of telomeres against GSK3*β*-primed FBW7-mediated TPP1 degradation [35,213]. Thus, to the emerged molecular target of TPP1 accelerated turnover, TELODIN prevents chronic stress induced premature pulmonary ageing and fibrosis, highlighting an effective protection of TPP1 and thus telomere dysfunction as a novel approach for intervention into pulmonary fibrosis.

## 5. Conclusions and Perspectives

Pulmonary ageing and fibrosis occur in association with AEC2 stem cell senescence and SADD. AEC2 stem cell failure of differentiation to AEC1 cells in epithelial damage repair, and trans-differentiation to mesenchymal fibrogenic remodeling, are the two critical cellular processes mediated by telomere dysfunction. Downstream of stress-induced uncapping of telomeres or TGF-*β*-signaling repression of the telomerase *TERT* gene, AEC2 stem cell senescence drives alveolar epithelial fibrogenesis, representing a key molecular target for intervention. Preventing AEC2 stem cell senescence by promoting telomere capping integrity using TELODIN, or removing senescent cells by promoting apoptosis using senolytics, is emerging as a promising strategy in the studies of pulmonary fibrosis intervention.

More investigations are underway to detect and intervene in stress-induced premature pulmonary senescence and the early stage of fibrosis, informing molecular targeting strategies of prophylactic and therapeutic intervention. The contemporary strategies and technologies—such as sequencing single cell populations to differentiate gene profiling of cell senescence and differentiation statues, deciphering structural modifications post gene transcription, defining disordered molecular networks, and decoding denatured macromolecules and compromised protective complexes—will lead to more studies of interventions of alveolar stem cell senescence and SADD in pulmonary fibrogenesis. Dissecting cellular and molecular interplays will likely produce further insights into the development of premature cellular senescence and fibrosis for some prophylactic and therapeutic outcomes.

## Figures and Tables

**Figure 1 cells-11-00877-f001:**
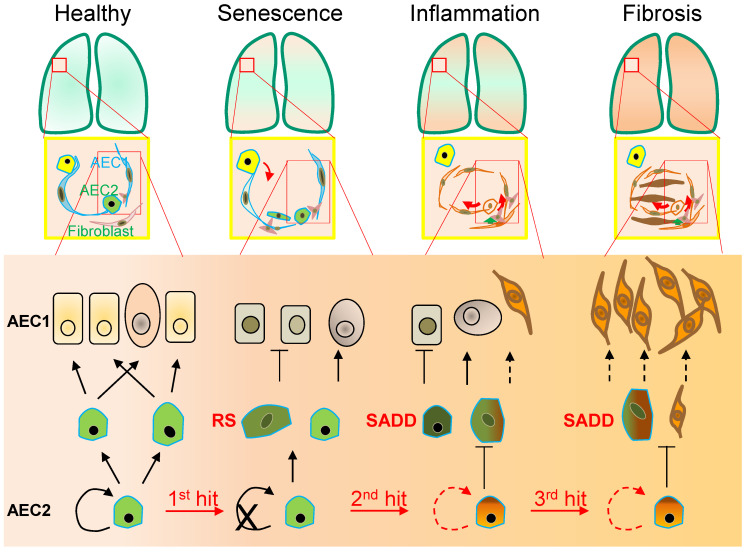
Alveolar monolayer squamous epithelial type 2 (AEC2) stem cell differentiation arrest and transdifferentiation disorder in pulmonary fibrosis. During pulmonary fibrogenesis, AEC2 stem cells are susceptible to stress assaults triggering telomeric DNA damage response (DDR) and replicative senescence and senescence-associated cease of the directional differentiation to alveolar monolayer squamous epithelial type 1 (AEC1) cells. Chronic stress induces senescent AEC2 stem cells to undergo transdifferentiation. The senescence-associated differentiation disorders (SADDs) contribute to myofibroblast proliferation under the condition of senescence-associated low grade inflammation (SALI).

**Figure 2 cells-11-00877-f002:**
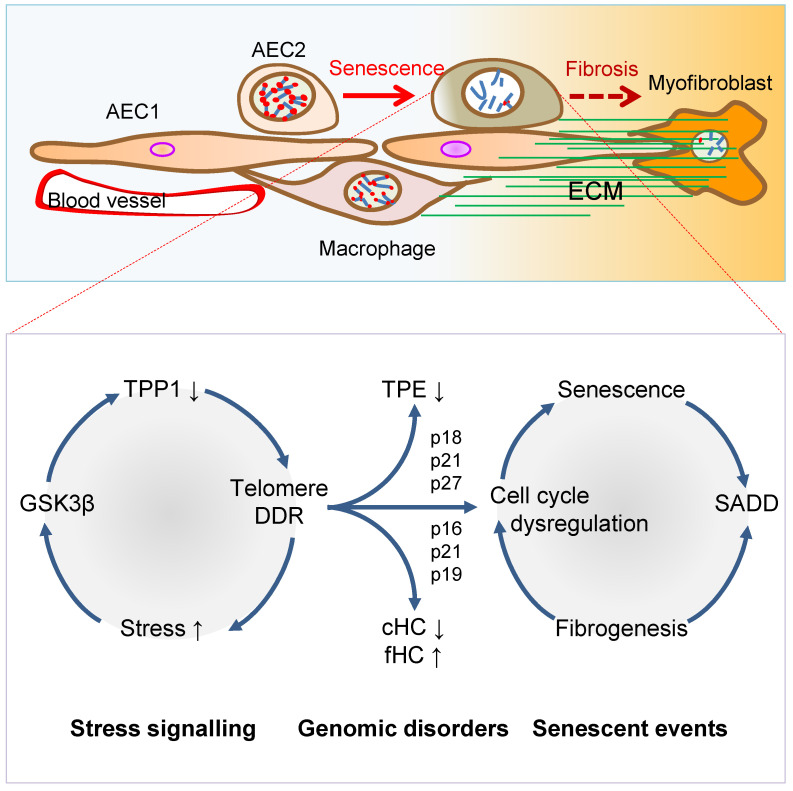
Mechanisms of AEC2 stem cell senescence and SADD. Cellular stress signalling triggers GSK3β-targeting of telomere shelterin complex, inducing the telomerase recruitment protein TPP1 phosphorylation, subjecting phosphorylated TPP1 multisite polyubiquitination and degradation, resulting in telomere uncapping. The telomere uncapping triggers telomeric DDR, resulting in activation of the cyclin-dependent protein kinase inhibitors and cell cycle deregulation through telomere position effect (TPE) and altered constitutive and facultative heterochromatins (cHC and fHC). Unresolved telomeric DNA repair and cell cycle arrest result in stem cell senescence and subsequent SADD including differentiation arrest and trans-differentiation underlying pulmonary fibrosis.

**Figure 3 cells-11-00877-f003:**
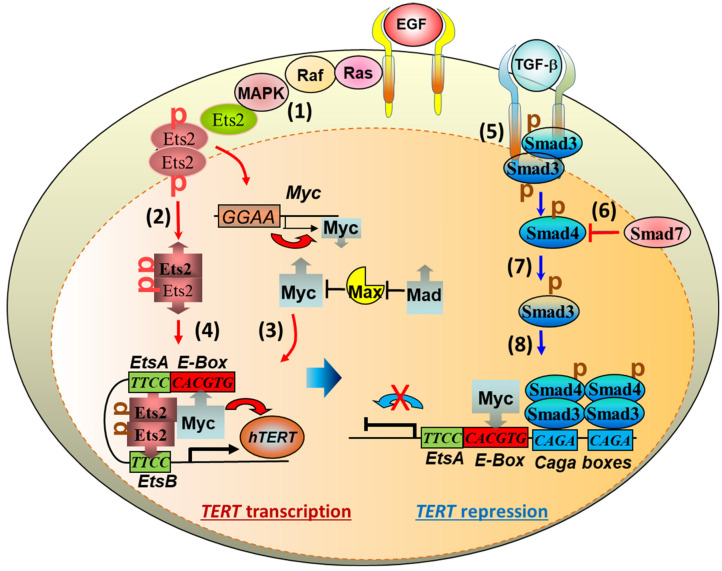
Intracellular signalling pathways of growth factors to the transcription factors and repressors in the regulation of telomerase reverse transcriptase (TERT) gene transcription. The TERT gene promoter assumes both active and repressive conformations under the molecular regulation of the MAP kinase and TGF-β signalling pathways, respectively. Epithelial growth factor (EGF) stimulated mitogenic signalling induces MAP kinase-mediated Ets2 transcription factor phosphorylation, nuclear retentions and dimerisation (**1**–**2**). Ets2 binds the CCTT element in the TERT and c-myc gene promoters, driving c-myc (**3**) and TERT (**4**) transcriptions in the upregulation of cell proliferative immortality. On the other hand, TGF-β activates TGF-*β* RII receptors by auto- and trans-phosphorylation, resulting in Smad3 phosphorylation and mobilization (**5**), which is regulated natively by Smad7 (**6**) and positively by Smad4 (**7**) in the Smad3 nuclear localization and action. Smad3 binds to the CAGA element in the TERT promoter to repress TERT gene transcription in pathological fibrogenesis such as pulmonary fibrosis (**8**).

**Figure 4 cells-11-00877-f004:**
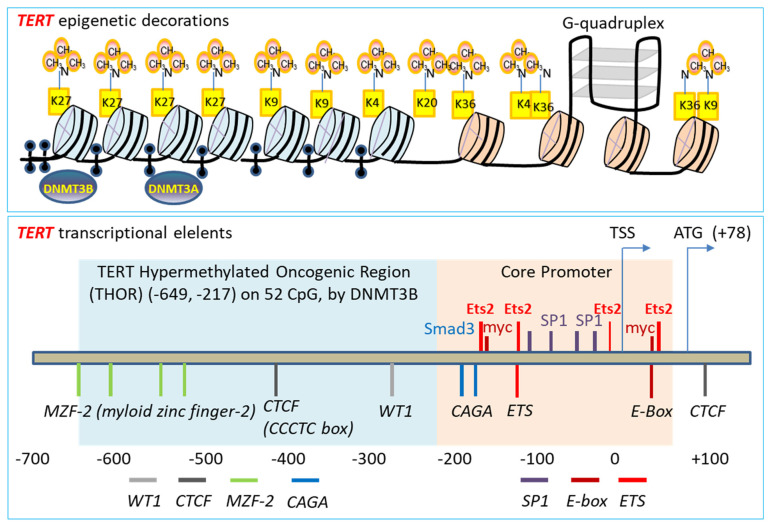
The regulatory network of TERT gene transcription. Top panel: Epigenetic regulatory organization of the TERT gene promoter repression involves trimethylation of the various lysine residues (K) on the nucleosome histone tails, methylation of C5 cytosine of CpG dinucleotides by DNA methyltransferase 3 alpha and beta (DNMT3A and 3B), and G-quadruplex. Bottom panel: TERT promoter DNA regulatory elements (italic), and engaged transcription factors and repressors (non-italic) in the positive (red) and negative (blue) regulations of TERT gene transcription. The scale labelled is for relative positions of promoter upstream regulatory DNA elements, but not in proportion.

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
