# Peer review of "Molecular Mechanisms of Alveolar Epithelial Stem Cell Senescence and Senescence-Associated Differentiation Disorders in Pulmonary Fibrosis"

_cells, 2022, doi:10.3390/cells11050877_

Round 1

Reviewer 1 Report

The present review by Hong et al., aims to summarize the “Molecular mechanisms of cellular senescence and senescence-associated differentiation disorders in pulmonary fibrosis”. This is an interesting issue for clinical research in pulmonary fibrosis, since senescence can be involved in this pathology: thus a critical assessment of novel findings could provide new routes for treatments.

Although interesting there are several concerns with the present manuscript that need to be addressed and I offer therefore the following:

  1. Similar very recent reviews address the same issue, this could lower the originality of the submitted manuscript. I suggest authors to ensure that there is no “copy” of same sections in the current literature. Moreover, please provide some compass for the whole review: this is necessary to improve readability.

  1. The manuscript must be emended in numerous sections/subsections to improve its quality. Since the sections 3-6 are key elements of this review, these should be reorganized and further integrated with section 2 for better readability. For example, lines 110-151 must be integrated with the whole manuscript.

  1. Extensive revisions of Figures and legends are also necessary. Figures are only presented in section 2 and absent in the other sections. Furthermore, Figure 3 must be carefully revised.

  1. It would help to have a Figure to summarize Section 5 and a Table for senolitic agents discussed in section 6.

  1. Although English usage and grammar is adequate, the authors will need to double-check the manuscript to remove odd constructions/sentences and to some spelling/typographical errors.

Author Response

  1. Similar very recent reviews address the same issue, this could lower the originality of the submitted manuscript. I suggest authors to ensure that there is no “copy” of same sections in the current literature. Moreover, please provide some compass for the whole review: this is necessary to improve readability.

Reply: We appreciated reviewer’s concerns on the originality of our paper. We would like to indicate that all descriptions are from our careful analysis of current literature available. These include the entire new concept of senescence-associated differentiation orders, and cutting edge discussions of telomere dysfunction mediates pulmonary senescence and fibrosis, which cannot be found in other lit reviews in terms of any similarity. In addition, on the mechanisms of different types of replicative senescence and molecular intervention are also original and literature citations are against current publications.

  1. The manuscript must be emended in numerous sections/subsections to improve its quality. Since the sections 3-6 are key elements of this review, these should be reorganized and further integrated with section 2 for better readability. For example, lines 110-151 must be integrated with the whole manuscript.

Reply: We appreciated reviewer’s concerns. We accordingly stay focused on molecular mechanisms of telomere sinalling and molecular intervention by removing the section 4 of virus-induced senescence and selectively amalgamated section 2 with section 1 introduction.

  1. Extensive revisions of Figures and legends are also necessary. Figures are only presented in section 2 and absent in the other sections. Furthermore, Figure 3 must be carefully revised.

 Reply: We appreciated reviewer’s comments. We have revised the figures and legends, by adding another figure (Figure 4).

  1. It would help to have a Figure to summarize Section 5 and a Table for senolitic agents discussed in section 6.

Reply: We appreciated reviewer’s comments. We have accordingly added a table for all senolytic agents (Table 1).

  1. Although English usage and grammar is adequate, the authors will need to double-check the manuscript to remove odd constructions/sentences and to some spelling/typographical errors.

Reply: We appreciated reviewer’s comments. We have edited the entire manuscript for English grammar and corrected typos throughout.

Reviewer 2 Report

In the review entitled “Molecular mechanisms of cellular senescence and senescence-associated differentiation disorders in pulmonary fibrosis”, Hong et al. summarize the studies investigating the roles of senescence in the development of lung fibrosis. Briefly, they describe the molecular mechanisms implicating epithelial senescence as a driving force of pulmonary fibrosis: i) Environmental stressors and therapeutic insults inducing unresolved DNA damages that leads to replicative senescence of epithelial AEC2 stem cells ii) Senescent AEC2 losing their capacity to give rise to AEC1 cells and instead transdifferentiate into mesenchymal-like cells fostering a pro-inflammatory environment iii) Molecular consequences of telomere dysfunction causing senescence and fibrosis. The authors end by presenting the therapeutic strategies to either prevent senescence or clear senescent cells to improve patients with pulmonary fibrosis. This manuscript will provide the reader with an updated overview of cellular senescence in lung fibrotic disorders, particularly molecular and cellular mechanistic models describing senescence as a driving force of lung fibrosis.

General comments : Senescence is a hot topic in the field of pulmonary fibrosis and therapeutic intervention via clearance of senescent cells hold great promises. The review is written correctly. The citations are accurate and up to date.  

  • The authors concentrate on the contribution of epithelial cells senescence in the physiopathology of lung fibrotic diseases. However, other lung compartments such as immune or mesenchymal cells present hallmark features of senescence in pulmonary fibrosis. Thus, to distinguish from others recent reviews on cellular senescence in lung fibrosis (e.g. https://doi.org/10.3390/ijms22126214), it may be important to precise that the review focus on alveolar epithelial senescence.
  • Before presenting the causal roles of telomere dysfunction in mediating pulmonary fibrosis and therapeutic interventions to limit senescence during fibrogenesis, the authors present two aspects of AEC2 cell senescence: molecular mechanisms of replicative senescence and its impact on differentiation to AEC1. However, other types of senescence such as Virus-Induced Senescence (VIS) or Oncogene Induced Senescence (OIS) are intermingled in the different paragraphs. In addition, I did not understand the aim of the SARS-Cov-2 paragraph, specifically the link with the paragraphs that precede. For the sake of clarity, I would suggest to present the different types of senescence, including the VIS observed after SARS-Cov-2 infection, in a dedicated paragraph (i.e. in the introduction) and thus, remove the SARS-Cov-2 paragraph. Then, I would expunge the other types of senescence and concentrate on presenting the molecular mechanisms of replicative senescence, the associated differentiation disorders, the roles of telomere dysfunction and the interventions.

Specific comments :

  • I would modify the title a little bit to clearly state the review focus on AEC2 replicative senescence and associated differentiation disorders.
  • Use of abbreviations which are close such as SADD, SALI make the arguments sometimes not easy to follow. If the authors could limit or simplify the abbreviations, that would enhance the readability of the review.
  • Figures are appropriate.

Author Response

Thus, to distinguish from others recent reviews on cellular senescence in lung fibrosis (e.g. https://doi.org/10.3390/ijms22126214), it may be important to precise that the review focus on alveolar epithelial senescence.

Reply: We appreciated reviewer’s concerns on a potential overlapping area. However, the review paper published in International Journal of Molecular Science was on cellular and subcellular mechanisms involving mitochondria, autophagy, cell cycle and inflammation. We have accordingly highlighted our focus on telomere dysfunction and TGF-beta signalling. These highlights We have  

I would suggest to present the different types of senescence, including the VIS observed after SARS-Cov-2 infection, in a dedicated paragraph (i.e. in the introduction) and thus, remove the SARS-Cov-2 paragraph. Then, I would expunge the other types of senescence and concentrate on presenting the molecular mechanisms of replicative senescence, the associated differentiation disorders, the roles of telomere dysfunction and the interventions.

Reply: We appreciated reviewer’s advice. We have accordingly removed the section 4 of virus-induced senescence and combined the section 2 with section 1.

Specific comments :

  • I would modify the title a little bit to clearly state the review focus on AEC2 replicative senescence and associated differentiation disorders.

Reply: We appreciated reviewer’s advice. We have accordingly modified the tittle and abstract indicating our focus on AEC2 cell replicative senescence.

  • Use of abbreviations which are close such as SADD, SALI make the arguments sometimes not easy to follow. If the authors could limit or simplify the abbreviations, that would enhance the readability of the review.

Reply: We appreciated reviewer’s advice. We have removed the use of SAHF and reduced the use of SALI.

  • Figures are appropriate.

Reply: Thank you very much.

Round 2

Reviewer 1 Report

The manuscript has been sufficiently improved to warrant publication in Cells

Author Response

Thank you very much. Your encouragement is very much appreciated. 

We have further checked English expression grammars and done the corrections.